# Sharing Art as a Daily Resistance Strategy in Madrid during the 2020 Lockdown: 50 Days of Collective Experience at the Plaza de San Bernardo

Laia Falcón 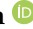

Departamento de Ciencias de la Comunicación Aplicada, Media Faculty, Complutense University of Madrid, Avenida Complutense s/n, 28014 Madrid, Spain; lfalcon@ucm.es

**Abstract:** The manner in which individuals worldwide shared art during the most challenging months of the COVID-19 pandemic stands as one of the most significant instances of creative social resistance in recent history. As a collective tool of resistance against emotional trauma, and as a means to foster a sense of community and well-being, the study of this phenomenon offers a compelling avenue for research into creativity and its social functions. This paper presents a descriptive case study of a successful 50-day collective experience within a neighborhood community in Madrid, Spain, during a period when the city, as a notably exceptional case study for research, bore one of the heaviest burdens of COVID-19 in the world. Data were gathered through in-depth personal interviews and direct observations. Applying a connected approach drawing on the fields of the Sociology of Art and Media Studies, three key findings emerge: (1) participants emphasized shared *live* artistic performances as the primary catalyst for fostering a sense of community, collective resilience, and overall well-being; (2) their sense of togetherness was further bolstered by digital and media support, as recordings of *live* performances were shared with loved ones living elsewhere, as well as with journalists and on social networks. This network of communication played a pivotal role in connecting individuals; (3) the combined efforts of both initiatives contributed to the development of a more positive individual and shared narrative surrounding the crisis.

**Keywords:** social resistance; community organization; art; well-being; digitalization; memory; COVID-19





## 1. Introduction

On 11 March 2020, the World Health Organization (WHO) declared the official onset of the COVID-19 pandemic, following the identification of 125,000 cases in 118 countries (WHO 2020). This momentous declaration marked the beginning of an unprecedented global crisis, characterized not only by the overwhelming loss of life and health but also by the emergence of a profound worldwide emotional trauma (Lahav 2020). Two months later, on 14 May 2020, the WHO sounded the alarm regarding the severe impact of the pandemic and its associated repercussions on mental health (Morales-Vives et al. 2020, p. 1). This unprecedented situation led to a surge in severe distress, anxiety, and depressive symptoms, as initially observed in early research conducted in various countries, including in China (Wang et al. 2020), Italy (Moccia et al. 2020), Spain (Morales-Vives et al. 2020), and the United States (Czeisler et al. 2020). Subsequent long-term studies have further corroborated these findings (Nussbaumer-Streit et al. 2020; Cheng et al. 2021; Bartrés-Faz et al. 2021; Eklund et al. 2022; Hoppen 2023). This global phenomenon has ushered humanity into an unprecedented historical chapter, characterized by what has been described as "the most significant worldwide stress-inducing event. It presents an unparalleled global challenge, particularly in the realm of mental healthcare. These challenges are intricately linked to the morbidity and mortality brought about by the disease itself and the various mitigation

measures implemented, including the profound impact of physical distancing and stay-at-home orders" (Czeisler et al. 2020, p. 1).

In response to the profound emotional toll of the pandemic, various forms of spontaneous artistic expression began to surface across different contexts early in the lockdown period. As underlined by specific research on the Sociology of Music during the Pandemic (Lange and Sun 2021; Jabłońska 2021; Vandenberg et al. 2021; Lettera 2022), among these creative outlets, music emerged as one of the most fervently embraced means for coping (Ferreri et al. 2021), emotional self-regulation (Centeno et al. 2021), and confronting feelings of loss, fear, and nostalgia (Yeung 2020). In the context of lockdown and the Digital Era, online sharing became the most logical and immediate response. It played a pivotal role in facilitating this creative process. Adapting to this novel and unfamiliar environment, orchestras, bands, choirs, companies, and individual performers spanning various genres and from across the globe came together to share their lockdown performances on various media platforms and social networks (Fraser et al. 2021). In some instances, this sharing of performances served as a means to sustain a presence and shed light on the significant economic and symbolic repercussions of the pandemic, also within professions closely tied to *live* arts, notably those within the music industry, which bore the brunt of the stay-at-home restrictions. However, the primary and overarching motivation behind these shared performances was to foster a sense of collective entertainment and encouragement: singers, instrumentalists, dancers, and actors, representing a diverse spectrum of ages and levels of popularity, united in their efforts to uplift and engage the global audience, transcending boundaries and sharing messages of resistance and strength to countless individuals. It is noteworthy to observe how numerous art celebrities actively embraced this trend and played an essential role as mainstream references and role models in its development and impact. As the world found itself confined to their homes, these stars, such as Lady Gaga, Yo-Yo Ma, or Meryl Streep, transcended their celebrity status. They too appeared from their computer cameras, often in ordinary clothing, just like the rest of us. In these moments, they exhibited vulnerability, sharing in the collective uncertainty, fear, and mourning that gripped society. Yet, they remained openly committed to following the guidance of health authorities and contributing to the reinforcement of calm and hope during this challenging period. Through the sharing of posts, engaging in chats, and other forms of online interaction, the audience's response to these artistic materials evolved into a crucial form of social engagement. Sociological, artistic, and media research have identified these processes as noteworthy instances of collective resilience against isolation and fear in the Digital Era. In the face of a global crisis, individuals utilized art, media, and digital tools to foster a sense of closeness and shared awareness, effectively combating feelings of isolation and fear (Montejano Hernández and Rojas Pedraza 2020; Yeung 2020; Miller and McDonald 2020; Granot et al. 2021; Centeno et al. 2021; Ferreri et al. 2021; Fraser et al. 2021; Martínez-Castilla et al. 2021; Alvarez-Cueva 2022; Feen-Calligan et al. 2023).

In addition to the online strategies, the *live* artistic activities that unfolded from the windows of quiet streets also held great significance for research and our understanding of this significant moment in history. These impromptu displays of creativity likely stemmed from another powerful societal urge. In countries like Spain and Italy, which were among the first to face the severity of the pandemic after China, individuals from various artistic backgrounds, both professional and amateurs, initiated spontaneous open-air performances from their windows and balconies right from the outset of the most stringent lockdown measures. These alternative activities, with musicians often at the forefront, were not only driven by the desire for collective entertainment, encouragement, and connection but also evolved into spontaneous conduits for fostering a deeper sense of collective interaction and well-being. From the reassuring proximity of their windows and the tangible sense of "shared breath" and "physical coexistence" that lie at the heart of *live* artistic performances (Riccioni 2021, p. 437), these periodic activities organized by neighbors—including small concerts, recitals for children, and open-air film projections on walls—assumed a key role in helping many citizens cope with grief and isolation. They played an essential part in

nurturing social identity and fostering a profound sense of community, a sentiment that proved indispensable for resilience, psychological well-being, and overall mental health during the challenging period of lockdown (Drury et al. 2020).

It is crucial to emphasize that online activities also played a significant role in shaping this sense of social identity, often operating on a global scale. However, certain local, *live*, periodic interactions within neighborhoods (with a very limited geographic scope) forged an even stronger and more potent sense of community, particularly concerning emotional subjective well-being. This was evidenced by research on collective applause for healthcare workers, highlighting the profound impact of these neighborhood gatherings (Canto and Vallejo-Martín 2021; Manthorpe et al. 2022; Zlobina and Dávila 2022). Moreover, the *live* artistic activities performed from windows and balconies represented an equally remarkable dimension in the realm of social resistance against the shared adversaries of fear, loss, and isolation. These activities offer an intriguing avenue for research into collective creativity and its various social functions during times of crisis.

The case of Spain stands out as highly significant in the context of these *live* artistic resistance activities. What makes Spain's situation particularly valuable for research is the convergence of two pivotal factors: firstly, Spain (and notably Madrid) endured one of the most severe initial waves of the pandemic during its early months (Bartrés-Faz et al. 2021), and consequently faced some of the most stringent confinement measures; on the other hand, Spain, from its deeply ingrained social habits and a strong desire for interpersonal interaction, witnessed a very high level of engagement among its population in the collective strategies aimed at fostering *live* daily contact. This intensity and the effectiveness of these strategies have been highlighted in studies like Morales-Vives et al. (2020). One of the most widely embraced strategies by the Spanish population was the collective applause in honor of frontline healthcare and social workers (Canto and Vallejo-Martín 2021; Zlobina and Dávila 2022). This international phenomenon took on exceptional significance in Spain, where it continued every day for 65 consecutive days, a frequency that far exceeded other similar periodic schemes adopted in various countries, such as the weekly routine in the Netherlands and the United Kingdom, for example. Furthermore, this practice was found to be "strongly associated with heightened emotional synchrony and, indirectly, with the prediction of self-reported preventive behavior" (Zlobina and Dávila 2022, p. 1). Its social strength became so powerful that from April 27th certain political right-wing parties encouraged the use of it as a daily collective date for expressing opposition to the Government. In addition to the widespread applause (and notably not so linked to political uses), collective artistic activities performed from windows and balconies emerged as a central coping strategy for many neighborhoods. Participants enthusiastically underscored these activities as a primary means of promoting well-being and fostering resilience (Cabedo-Mas et al. 2021).

This research aims to make a contribution to the expanding body of literature that examines collective responses during the COVID-19 pandemic from an international comparative perspective (Brooks et al. 2020a, 2020b; Drury et al. 2020; Yeung 2020; Granot et al. 2021; Centeno et al. 2021; Fraser et al. 2021; Hoppen 2023), and in the specific case of Spain (Bartrés-Faz et al. 2021; Morales-Vives et al. 2020; Cabedo-Mas et al. 2021; Canto and Vallejo-Martín 2021; Martínez-Castilla et al. 2021; Postigo-Zegarra et al. 2021; Alvarez-Cueva 2022; Zlobina and Dávila 2022). To achieve this objective, this paper centers its investigation on a specific case study: the transformation of a neighborhood consisting of 13 buildings located in the heart of Madrid into a cohesive community. During the most challenging phase of the lockdown, from 14 March to 1 May 2020, this community applied the power of art as a means of mutual emotional support and well-being to counter the effects of isolation, fear, and loss. The urban block is situated in the Chamberí neighborhood, specifically around the Glorieta de Ruíz Jiménez, commonly known as the "Plaza de San Bernardo" by most of Madrid's residents. During this lockdown period, this area was home to a diverse population comprising households of families (many with children), university students, and young professionals sharing apartments, as well as elderly individuals, some of whom

lived alone. Additionally, the block was host during working hours to the employees of the only three establishments open at the time: two pharmacies and a small convenience store. Before the lockdown, it was rare for most of these residents and workers in the three shops to engage in conversation with each other. However, during the lockdown period they underwent a transformation into a dynamic community of resilience, primarily through the artistic expressions they shared from their open windows. This included a daily performance by a professional lyric singer, weekend film sessions projected onto the walls of the buildings, the weekly creation of collaborative pop singing videos, themed carnival days for the children and young adults, and the commemoration of the national Day of the Book. Initially, arising spontaneously, these shard artistic activities were quickly organized into a variety of systematic routines and strategies. Within this community, they served several crucial purposes: they facilitated daily social interactions from windows and balconies, provided a structural framework for leisure with fixed schedules and breaks, and, above all, emerged as a vital collective instrument for withstanding the profound challenges of isolation, uncertainty, and grief during the most devastating phase of the pandemic in Madrid (at the time, one of the hardest-hit regions globally). The primary specific objectives of this study are as follows:

1.  to determine whether members of this community who engaged in intensive *live* artistic interactions had a distinct perception of the emotional and social aspects of the lockdown, in comparison to individuals residing in neighborhoods without such *live* artistic interactions;
2.  to assess whether this *live* interaction was perceived as more preferable than online interactions, even when established with randomly selected ("non-chosen") citizens with whom there had been no prior contact until the lockdown.

Two central hypotheses were to be tested. Hypothesis 1: *live* artistic activities shared in neighborhoods served as potent catalysts for cohesion, emotional resilience, and resistance during the crisis. Hypothesis 2: *live* interactions, even when initiated by citizens with whom there had been no prior contact until the crisis, were favored and more effective in terms of emotional closeness within the local community.

Drawing on direct observation, a comparative analysis of 60 in-depth personal interviews (30 conducted with members of this community, 30 with citizens who were locked down in other areas of the city), and an integrated approach that bridges the perspectives of the Sociology of Art and Media Studies, this study discovered three prominent patterns. Firstly, a notable finding is that all the participants who engaged in collective artistic sharing consistently associated their lockdown experience with a profound sense of community and closeness. In contrast, this aspect was either vaguely mentioned or not mentioned at all by citizens who did not take part in such activities. Moreover, participants emphasized, without prompting, that the sharing of artistic expressions stood out as one of the most significant factors in their personal resilience against trauma and the overall well-being of the community: these artistic activities offered daily opportunities for non-digital social interactions, which were greatly appreciated even within the confines of window-to-window safety distancing. They played a crucial role in structuring time, helping residents mark the hours of the day, the progression of the weeks, and the distinction between workdays and leisure days. Additionally, they served as a key mechanism for generating a collective sense of encouragement (a sentiment that resonated strongly among both the residents of the building and the employees in the local shops) and, in certain cases, as a way to share mourning and healing processes. Secondly, this study underscores the substantial role played by digital and media assistance in supporting communal life within this community. The sharing of video recordings of these activities fostered a novel routine of external contact with loved ones residing elsewhere and forged a network of communication with the media, which evolved into a crucial lifeline for the community's sense of collective resilience; thirdly, the harmonious collaboration between these two initiatives contributed to the development of a more favorable individual and shared narrative concerning the crisis.

Hypothesis 1 is, therefore, confirmed but not Hypothesis 2: while participants emphasized the importance of *live* interaction (even with individuals they had not known before the crisis) as a critical factor for well-being and resistance, they also emphasized the effectiveness of online and media interaction in safeguarding emotional well-being and fostering a sense of local togetherness.

These findings offer valuable insights for empirical sociological research during this pivotal period in our recent history, by detecting and analyzing the ways in which individuals and communities employed art as a tool for resilience and well-being during such times of crisis.

## 2. Materials and Methods

The core of this research is the direct observation conducted during a specific timeframe, from 13 March to 1 May 2020, a period marked by two significant events: the announcement of the lockdown in Spain, and the subsequent easing of restrictions, allowing the population to venture outside during designated hours of the day. This study's foundation is further consolidated by a qualitative comparative analysis involving 30 in-depth interviews with members of the aforementioned community. Additionally, 30 more in-depth personal interviews were conducted with individuals whose experiences of the lockdown occurred in other areas of the city. The daily direct observation encompassed a total of 74 residents who participated either as performers or as engaged members of the audience in the artistic activities held within the Plaza de San Bernardo community. Tables 1 and 2 provide detailed descriptions of their respective profiles.

**Table 1.** Profile of 74 community members (part I).

| Age | Gender | Type of Membership |
|---|---|---|
| children 0–3: 3 members | female: 39 | inhabitants [1]: 61 |
| children 4–10: 7 | male: 32 | shop workers [2]: 12 |
| teenagers 11–17: 3 | other: 1 | |
| adults 18–29: 16 | | |
| adults 30–59: 27 | | |
| adults 60–69: 5 | | |
| adults older than 70: 12 | | |

[1] One of the inhabitants (working in a hospital) was allowed to leave the neighborhood during working hours. The remaining adults only had permission to leave their homes to go to the nearest pharmacy or food store, and to deposit garbage in street containers. In Spain, children were asked to stay at home during this period, except those with special needs (such as diagnosed autism, for instance). [2] None of the workers at the pharmacies or the convenience store lived in the neighborhood. They were present in the neighborhood only during shop opening hours and returned to their houses in other parts of the city after work and remained there on non-working days.

**Table 2.** Profile of 74 community members (part II).

| Lockdown Situation | Health Situation Regarding COVID-19 | Country of Origin |
|---|---|---|
| in family with children: 30 | severe hospitalization: 2 | Spain: 71 |
| in couple with no children: 14 | not hospitalized: 72 | China: 1 |
| sharing an apartment: 14 | | Ecuador: 1 |
| living alone [1]: 4 | | Argentina: 1 |
| working in the shops: 12 | | |

[1] Three of the members living alone were able to go out for shopping; the other one did not leave the house during the whole period, receiving assistance from neighbors or the shop workers for food purchases and the taking-out of garbage.

While this initial broad reference group is diverse in several aspects, it is important to note that they do not serve as a representative sample of the Madrid population. The historical circumstances leading to this episode were marked by dramatic events, and the primary research focus regarding this group of neighbors was not to obtain a demographic sample. This sample was not strategically designed for the research but rather shaped by

the economic and social realities of these specific citizens. They primarily consisted of individuals from a broad middle-class demographic who happened to reside in close proximity within these 13 buildings during the lockdown period. Their collective engagement in artistic activities emerged as an act of communal resistance during this time. Among the 74 active members of this community was the author of this paper. At the time, I was a resident of one of these buildings, and I also happened to be the professional lyric singer who began performing from one of the windows during the first week of the lockdown, starting 21 March. While in other types of research such personal involvement might pose challenges in terms of data collection, emotional distance, or objectivity, in this unique and dramatic setting it proved to be a key factor in launching data collection and, in a way, rendered the research feasible, as I will elaborate on in the following sections.

From the group of 74 members, 30 were selected for the initial phase of in-depth personal interviews, designed as Interview Group A (IGA). The criteria for selection focused only on adults and aiming to create a representative sample of the community with regard to gender, age, occupation, and their circumstances during the lockdown. Detailed profiles of the chosen participants are presented in Tables 3 and 4.

**Table 3.** Interview Group A (IGA): profile of the 30 participants for in-depth personal interviews (part I).

| Age | Gender | Type of Membership |
| --- | --- | --- |
| adults 18–29: 4 | female: 15 | inhabitants: 24 |
| adults 30–59: 20 | male: 14 | shop workers: 6 |
| adults 60–69: 3 | other: 1 | |
| adults older than 70: 3 | | |

**Table 4.** IGA: profile of the 30 community participants for in-depth personal interviews (part II).

| Lockdown Situation | Health Situation Regarding COVID-19 | Country of Origin |
| --- | --- | --- |
| in family with children: 10 | severe hospitalization: 2 | Spain: 27 |
| in couple with no children: 6 | not hospitalized: 28 | China: 1 |
| sharing an apartment: 4 | | Ecuador: 1 |
| living alone: 4 | | Argentina: 1 |
| working in the shops: 6 | | |

The interviews extensively addressed the experiences of children and teenagers, albeit through the accounts provided by their parents, and not from the children or adolescents themselves. It is worth noting that participants who lived alone, individuals who constituted severe hospitalization cases, and those from other countries, were intentionally oversampled. The experiences of these specific participants were considered particularly valuable in gaining a deeper understanding of the diverse realities within the community.

It should be noted that the recruitment of participants began in a rather organic and meaningful manner, largely initiated by the participants themselves. The extraordinary circumstances we all found ourselves in played a key role in this process. Being recognized as "the opera singer at the window", as many of them referred to me at the time, sparked interest and prompted as many as 15 neighbors to proactively reach out, eager to share their own stories. In 4 instances, they actively sought out my institutional email address or my Facebook account to establish contact. In the remaining 11 cases, neighbors approached me directly, often starting with the question, "Excuse me, are you the opera singer in the window?" These encounters typically occurred by chance, whether at the local convenience store or the nearby pharmacy. Furthermore, the singing from the window inspired two other neighbors to reach out to the media, specifically the national newspaper ABC and Madrid's public television channel Telemadrid, both of which contacted me on 26 May. This unexpected media attention not only significantly facilitated the recruitment of participants

but, as I will describe in subsequent sections, it also emerged as a significant factor in fostering increased social interaction within the community. Given that many of us were previously unacquainted with one another, these digital and media tools proved invaluable in establishing connections within the community.

The 30 interviews comprising Interview Group A (IGA) were conducted during the second and third weeks of May 2020. To provide a comparative reference for gaining a deeper understanding of the community's behavior, an additional 30 in-depth personal interviews were conducted with individuals who had experienced the lockdown in other parts of the city. These interviews, forming the Interview Group B (IGB), were gathered during the third and fourth weeks of May and the first and second weeks of June 2020. As shown in Tables 5 and 6, deliberate efforts were made to select participants that closely matched, as far as possible within the constraints of the lockdown context, the demographics of gender, age, and situation during the lockdown, and country of origin of IGA members, in order to establish an equivalent sample.

**Table 5.** Interview Group B (IGB): profile of the 30 participants from outside the target community for the second collection of in-depth personal interviews (part I).

| Age | Gender |
|---|---|
| adults 18–29: 9 | female: 14 |
| adults 30–59: 15 | male: 14 |
| adults 60–69: 3 | other: 2 |
| adults older than 70: 3 | |

**Table 6.** IGB: profile of the 30 participants from outside the target community for the in-depth personal interviews (part II).

| Lockdown Situation | Health Situation Regarding COVID-19 | Country of Origin |
|---|---|---|
| in family with children: 15 | severe hospitalization: 3 | Spain: 27 |
| in couple with no children: 7 | not hospitalized: 27 | Russia: 1 |
| sharing an apartment: 4 | | Portugal: 1 |
| living alone: 4 | | Chile: 1 |

Members of both IGA and IGB were asked about their personal experiences during the lockdown. The interviews covered a range of aspects, such as their health, emotional well-being, any changes in job circumstances resulting from the COVID-19 crisis, significant episodes they encountered during the lockdown, the care and situation of loved ones (with particular focus on elderly relatives and children), and how they organized their time. Two important topics were intentionally omitted from the explicit questions: the participation in, and sharing of, artistic activities, as well as any sense of belonging to a community during the lockdown. As I will elaborate on in the following section, it was important for this research that both these aspects, if mentioned, be spontaneously brought by all the participants who had engaged in collective artistic activities.

The interviews were carried out by phone, this being the method preferred by some participants, particularly the elderly. Each interview lasted between 30 and 60 min. All participants provided informed consent for their involvement in this study. In order to maintain confidentiality, they are identified by coded references that encompass elements of their gender identity, age, occupation, original country, and situation during the lockdown.

## 3. Results

The results revealed three principal findings: firstly, the fact that all those who engaged in artistic neighborhood activities, including all participants in IGA and one from IGB, emphasized the significance of concepts associated with collective sharing and a sense of community. They often used expressions like "becoming a strong group", "closeness",

and "resisting together" when describing their personal lockdown experiences. In stark contrast, these concepts were either vaguely mentioned or not mentioned at all by the remaining 29 participants in IGB who did not take part in such activities. The second notable finding involves digital and media assistance in supporting this *live* artistic communal existence. The regular online sharing of video recordings of these activities not only facilitated communication with loved ones living elsewhere but, particularly in the case of the Plaza de San Bernardo neighbors (IGA), also established a web of interaction with the media. This media connection proved to be pivotal for the community and its collective resilience. The third finding emerges from the unique synthesis of the two preceding ones: all participants who engaged in collective artistic activities spontaneously highlighted these activities as among the most significant tools for constructing an everyday narrative of resistance, well-being, and coping with grief.

In terms of the central hypotheses and primary objectives of this research, it is worth underlining that:

1.　members of the community who engage in *live* artistic activities tend to experience a heightened sense of cohesion and a greater awareness of the significance of *live* social interaction and support as effective tools for resisting crises.
2.　this *live* interaction is widely regarded as a key factor for well-being and resilience, even when established with individuals who had no prior contact before the lockdown. Furthermore, online and media interaction is also highlighted as a highly efficient tool in promoting well-being, storytelling, and resistance against crises.

Thus, Hypothesis 1 is confirmed but Hypothesis 2 is not: live interaction is highly valued, but it does not supplant online and media interaction, rather it complements it.

### 3.1. Sense of Safety and Closeness through Art: The Significance and Awareness of Building a Resilient Community

It is particularly noteworthy that when participants describe their personal experiences during lockdown, all those who engaged in artistic neighborhood activities emphasize the significance of collective sharing among neighbors. The specific phrases "becoming a close-knit group" (akin to the Spanish expression "hacer piña"), "closeness", and "resisting together" were used by all 30 participants from the San Bernardo community (IGA), as well as by one participant who experienced lockdown in another part of the city (IGB), where similar artistic activities were also conducted among neighbors.

### 3.1.1. Live Artistic Interaction and Sense of Togetherness

All IGA participants and one IGB participant (the only one taking part in social art activities) underline a feeling of unity closely tied to artistic collaboration and the experience of real-time coexistence, as opposed to online social interaction. All IGA participants connect their accounts of the *live* artistic activities with the daily collective applause. Such descriptions of the shared applause tend to be extensive and more emotionally charged in the testimonies of those who actively participated in *live* artistic activities (all members of IGA and one of IGB). These moments were described as catalysts for a profound sense of community marked by intense inner meaning and emotional depth. In contrast to these accounts, among the 29 participants who did not engage in *live* shared artistic activities (29 participants of IGB), 21 of them acknowledged the significance of the initial phase of collective applause (until 27 April, as further elaborated) in providing emotional support. However, none of them depict it with the vivid expressions of shared identity (such as "behaving as a strong group" or "togetherness"), but rather as an important daily ritual.

When considering the connection between both sorts of *live* interaction activities, it is noteworthy that some participants in IGA specifically emphasize an important distinction: the artistic interactions never became "entangled" with other dimensions, particularly political agendas. To provide context, it is important to mention that from 27 April, the far-right VOX party began advocating the gatherings for collective applause as a daily opportunity for anti-Government protests, often involving noise-making with pots and

pans. Four members of IGA and five from IGB supported this strategy. In contrast to the reasons and sentiments expressed by them, the shift in the use of the daily communal applause towards a new direction caused emotional discomfort for a significant number of participants (26 in IGA, 25 in IGB): the reasons articulated in their responses reflect significant insights into how *live* shared activities contributed to a sense of community, and to strategies of well-being and resilience. These accounts also highlight a distinct perception between the evolving direction of the shared applause and the consistent use of *live* artistic activities. The initial phase of collective applause and the *live* artistic activities are recognized as key factors in fostering a sense of community, while the second phase of the applause, now mixed up with the pots and pans protest, elicits two contrasting interpretations: those participants supporting the pots and pans protest view the noise-making as a form of resistance against perceived injustice (four members of IGA and five from IGB), making it the catalyst and primary symbol of this specific resistance community. These individuals are those who disagreed with the "official" versions and/or the adopted restrictions; in contrast, those participants who did not support the pots and pans protest (26 in IGA, 25 in IGB) perceive it to be disruptive to the feeling of togetherness, the sense of community, and to the shared resistance against fear and grief.

### 3.1.2. Live Artistic Activities and a Comforting Sense of Perception and Collective Organization of Time

One of the key characteristics shared by both modes of *live* interaction (shared applause and engaging in artistic activities) lies in their structured and recurring scheduling. This holds particular significance for all participants, with those engaging in artistic activities (IGA) emphasizing its increased importance. These events serve as vital occasions for maintaining daily social connections. Even when separated by constraints of physical distance, this framework allows for brief interactions with neighbors, confirming their presence, and collectively confronting the challenges of our social landscape. It is noteworthy that many IGA participants underscore the value of these shared schedules, viewing them as a very positive contributor to well-being and daily motivation. Indeed, such descriptions take on added significance when contrasted with a notable shared tendency among participants who did not engage in *live* artistic activities (29 participants of IGB). Of these, 24 highlight the absence of any strategy to infuse meaning into the passing of the days.

### 3.2. Media as Intermediary

In this study case, *live* responses and *live* communal spaces played a pivotal role in initiating and sustaining *live* art activities. In the Plaza de San Bernardo community, artistic engagement took its first steps on Saturday 21 March at 18:30, with a spontaneous lyrical singing performance of the aria "S'altro che lacrime" from Mozart's "La Clemenza di Tito". Yet, what transformed this individual spontaneous act into something more profound was the immediate response from the neighborhood. On hearing the voice echoing through the empty streets, several neighbors opened their windows and expressed their support for the music with smiles and similar gestures. In response to the applause and enthusiastic calls for an encore, other neighbors ventured out to discover the source of the commotion. The second and final aria, "O mio babbino caro" from Puccini's "Gianni Schicchi", followed suit. The collective response was one of jubilance, prompting numerous neighbors to shout out words of appreciation and interaction from their windows: "thank you!", "celebrating our first week!", "how beautiful!", "hello there!", "you're cheering us up!", "take care!", and "does anyone need help?" The same sequence was repeated the following day at the same time. This time the performance featured the leading melody of Offenbach's duet, "Barcarole" of "Les Contes d'Hoffmann", and again Mozart's "S'altro che lacrime". This served as a heartfelt celebration of the fact that we had successfully navigated the first nine days of lockdown. Moreover, it signaled our optimism that we might be drawing closer to the end of this challenging period, especially since the initial announcement of lockdown had indicated a duration of two weeks. Initially, this singing plan was for just two days

and no more. But the following Monday, at 18:35 when the performance had not yet begun, the neighbors began to call out from their windows for the singing to begin. There arose an impromptu chorus of "it's time to sing, it's time to sing" (¡*que cante, que cante!* in Spanish), and so, from that moment on, the lyric performances took place every evening at 18:30 for the next 47 days. This is how this specific community solidified its initial artistic tradition through *live* interaction, which soon inspired other creative endeavors. The younger adults took the initiative to organize weekly film screenings on one of the building facades. They also started a weekly project in which a song was selected every Monday, and neighbors of all ages were invited to record themselves dancing or engaging in playback performances during that week. This collaborative effort resulted in the creation of a shared piece. In addition to these activities, some families with children set out to create new carnival outfits almost daily for their little ones to "attend" each "opera concert" in style. The collective engagement in these activities was further reinforced by the daily participation of the pharmacy and convenience store workers who stood at the doorways as a symbol of shared encouragement. Even occasional observers like police officers and taxi drivers would frequently pause on their routes for a few minutes to listen in to the performances and express their appreciation. Taking inspiration from certain residents who, during first week of lockdown, displayed on their balconies banners supporting the public health system, a heartwarming trend emerged on 30 March with the appearance of new banners expressing gratitude for the music performances, with a total of five added in subsequent weeks. The occasional brief gatherings of neighbors at the three nearby shops became a platform to foster and promote these communal dynamics. At these informal gatherings, neighbors would extend their appreciation to the performers, discuss the impact of such artistic contributions, and even make specific musical requests for the upcoming sessions. Besides the physical interactions described earlier, this case study also underlines the key role played by digital and media platforms in fostering communal coexistence. Four key processes stand out:

1.  The compilation of video recordings captured by neighbors, and subsequently shared with journalists on social media platforms, from the outset of the window singing initiative formed a dynamic network of communication and collaboration that proved indispensable to this community. It played a central role in nurturing collective resilience and facilitating external connections. Witnessing "their" block's story featured in news broadcasts and newspapers, not only in Spain but also in international media outlets, was a source of great pride and celebration for the community: ABC (31 March 2020[1]), TeleMadrid (31 March 2020)[2], ImagenNoticias at Mexican Imagen Televisión (31 March 2020)[3], Diario.es (31 March 2020)[4], RNE (8 March 2020), SER (9 April 2020)[5], and SER (2 May 2020).

2.  As mentioned in the Materials and Methodology section, four participants reached out to me on Facebook or found my institutional email to express their gratitude for the music, thereby encouraging its continuation, and using it as a means to share their personal experiences.

3.  The sharing of the weekly dancing videos was organized around WhatsApp groups of neighbors.

4.  Streaming systems provided an opportunity for numerous community members to turn the daily window performance of lyric singing into a means of staying connected with friends, family, and work colleagues living elsewhere. Using their mobile devices and computers, they began to stream these daily moments with their loved ones. In some cases, this practice evolved into a platform for organizing highly intimate and significant virtual gatherings, including celebrations as well as a number of funeral services.

5.  Using the city hall's online resources, some neighbors successfully petitioned to have the fountain in the Plaza de San Bernardo turned off for 10 min every day at 18:30, so the lyric singing could be better heard in all the houses. This act of coordination and

cooperation was probably the most symbolic gesture within the broader collective movement of social resistance.

### 3.3. An Important Story to Tell: "Putting the Pieces Back Together"

The symbolism associated with the silencing of the fountain brings this analysis to a final observation, highlighting the way all participants in IGA emphasized the importance of crafting a narrative that was both informative and emotionally resonant. This collective mindset gave rise to four primary forms of action:

1.　some residents took the initiative to spontaneously contact media outlets to share the story of their community;
2.　the daily suspension of the fountain's operation for ten minutes, between 18:30 and 18:40 every evening;
3.　there was a poignant demand to stream certain lyric singing sessions, repurposing them as symbolic and intimate virtual funerals. This allowed friends and relatives who had just lost loved ones and were living elsewhere to participate in the grieving process;
4.　residents actively sought out opportunities to share beautiful or humorous anecdotes on a daily basis.

All participants of IGA consistently emphasized the significance of narrating this unique and profound experience, conveying three key ideas:

1.　the importance of openly discussing the unfolding events, including their emotional aspects, and skillfully framing the most significant occurrences with beauty and depth;
2.　the paramount importance of establishing a structured storytelling practice as a daily ritual for safeguarding well-being and fortitude;
3.　the importance of creating, sharing, and keeping "good memories", in order to enhance resilience and to preserve these memories for the future.

This collective response seems to be a direct result of the profound emphasis placed on *live* artistic interaction and the strong sense of community it fosters. Moreover, it has been significantly influenced by the fusion of these elements with the digital and media interactions. The persistence with which certain media outlets narrated experiences worldwide involving such *live* artistic activities, often focusing on themes that could provide emotional sustenance to the audience, further reinforced the participants' inner storytelling process. In this sense, it is noteworthy how the residents of the Plaza de San Bernardo community actively engaged with media outlets, exchanging and receiving news of their own case through social networks. What is particularly striking is how both forms of narrations (the media perspective and the accounts from the community participants) shared a common mood and style. For example, the words of Mexican journalist Yuriria Sierra (broadcast on *Imagen Televisión* on 31 March, and subsequently posted on YouTube under the hashtag "#Resistemexico") are a good example of this process:

> "*Facing social distance*, this *happens in Madrid every (e-ve-ry) evening. Please listen.* [The images show the first day of window lyric singing and the neighbors' reaction at the Plaza de San Bernardo, with the subtitle: "Spain. Soprano sings from her balcony in Madrid"] (. . .) *the intention is to forget for a few minutes the fear arising from the crisis. So many men, women and children sharing these acts of spontaneous solidarity, and beauty. I won't get tired of saying this: COVID is also bringing out the best in us*"—Yuriria Sierra, *Noticias con Yuriria Sierra*, 31 March 2020.[6]

This collective response seems to be a direct result of the profound emphasis placed on *live* artistic interaction and the strong sense of community it fosters. A few days after this broadcast, one of the Plaza de San Bernardo residents received a WhatsApp link to this content, and enthusiastically shared it with many of her neighbors through WhatsApp and Facebook, adding a cheerful comment: "Look! We are even on in Mexico!" This kind of "positive news" about their own community served to further reinforce their collective resilience and motivated them to actively seek out more beautiful memories to share. Active

participation in *live* artistic events as a means of creating cherished memories was a shared sentiment among all community members. They underlined these events as part of their collective identity, including when they were in the role of the audience. Noteworthy expressions of this process included children dressing up in carnival outfits to "attend" the "concerts", the creation of banners to celebrate the music, and the enthusiastic suggestions for specific arias or songs for the next lyric singing performances.

## 4. Discussion

This research offers significant insights for empirical sociological studies on one of the most profound, far-reaching, and challenging periods in recent history. It delves into how individuals and communities harnessed the power of art as a means of resistance and well-being during times of crisis.

The line underlined by Jabłońska (2021), Vandenberg et al. (2021), and Lettera (2022) offers significant insights about the specific role of music during the COVID-19 pandemic as a tool for protecting well-being, social resistance, and collective memories. In the specific case of Spain, as one of the hardest-hit countries during the first (and therefore most dramatic and uncertain) months of the pandemic, the results of this study case confirm the findings of Morales-Vives et al. (2020), Canto and Vallejo-Martín (2021), Cabedo-Mas et al. (2021), and Zlobina and Dávila (2022) on the key role played in Spain by the *live* interaction generated from balconies. While the Plaza de San Bernardo community may constitute a relatively small and non-random sample of citizens, and therefore by no means a representative sample of the Spanish population, it is important to note that the collective perception of participants within this group consistently underscores the profoundly positive impact of their communal engagements. This finding aligns with other research in the field. As noted by Morales-Vives et al. (2020, p. 8), in Spain the lockdown period saw a notable increase in neighborly interactions, thanks to communal events such as the daily applause for healthcare personnel, bingo games, and musical events. In particular, residents in the community of the Plaza de San Bernardo highlighted the remarkable transformation of a group of neighbors who had had little interaction before the crisis. They suddenly adopted an intense and supportive community style, which exemplifies the way crises can bring communities together.

This study shows the direct association between the effects of the applauses and the artistic balcony activities, as noted by Morales-Vives et al. (2020) and this recognition is shared by all IGA participants. In their testimonies, this association prompts participants to openly express feelings regarding the evolution of the applause tradition in Spain. These sentiments could serve as a valuable complement to the findings of Cabedo-Mas et al. (2021) and Zlobina and Dávila (2022): as observed by these authors, the participants in the first phase of the study initially perceived the applause ritual as a highly significant opportunity for social cohesion, fostering emotional connection and synchronization. However, this perception underwent a transformation for some participants due to the introduction of a new political dimension, starting on 27 April. It would be immensely valuable to apply the methodologies and representative approaches used by Cabedo-Mas et al. and Zlobina and Dávila to comprehensively assess this second phase of the applause tradition, to confirm and better understand the extent of this change in perception and its underlying factors.

Focusing on the societal function of communal artistic endeavors, particularly music, this paper corroborates the findings of Cabedo-Mas et al. In their research on the role of music during lockdown, these authors observed that a significant portion of the Spanish population regarded musical activities as a highly beneficial means to enhance well-being and bolster emotional resilience amidst isolation: in a nationwide survey comprising 1868 participants from all the country's autonomous communities, their research confirms that "31.1% of the sample actively engaged with these initiatives, and another 24.2% were not only aware of them but also actively initiated or took part in such endeavors" (Cabedo-Mas et al. 2021, pp. 3–4). The results of this paper's case study align with these findings, as all

participants who were involved in *live* musical activities emphasized their significance as essential tools for emotional resistance and well-being.

Regarding the utilization of music as a mechanism for resilience and inner storytelling, future research should examine the emergence of alternative *live* communal experiences within neighborhoods, exploring how communities collectively sought and exchanged the most effective methods and content for navigating crises. While the studies by Centeno et al. (2021), Martínez-Castilla et al. (2021), and Alvarez-Cueva (2022) primarily focus on various aspects of musical choices during lockdowns, such as increased consumption, music preferences, and emotional responses, they mostly center on online sharing and individual utilization. Nevertheless, their findings offer valuable insights that could prove essential if extrapolated to collective live-sharing activities.

Furthermore, future research examining the ongoing development of these practices during profound social crises, using longitudinal methodologies, will significantly enrich this area of study. The detection and analysis between different music genres and their specific social perceptions and uses in the public sphere during this sort of crisis may be also a notably interesting field of research. However, it is our hope that we will not have to face another situation like this in the future.

**Funding:** This research received no external funding.

**Institutional Review Board Statement:** The study was conducted in accordance with the Declaration of Helsinki, and approved by the Ethics Committee of Universidad Complutense (1-07-2008, approved on 11 June 2008).

**Informed Consent Statement:** Informed consent was obtained from all subjects involved in the study.

**Data Availability Statement:** The data presented in this study are available on request from the author. There is not publicly available due to privacy and ethical restrictions.

**Acknowledgments:** I thank participants for their outstanding generosity.

**Conflicts of Interest:** The author declares no conflict of interest.

## Notes

1     https://www.abc.es/espana/madrid/abci-opera-desde-balcon-contra-covid-19-teatro-mas-grande-estado-nunca-20200331033_video.html (accessed on 2 October 2023).

2     https://www.telemadrid.es/programas/120-minutos/musica-medicina-soprano-vecinos-Chamberi-2-2218298167{-}{-}2020033 1020638.html (accessed on 2 October 2023).

3     https://www.youtube.com/watch?v=fQ5M7q7grsE (accessed on 2 October 2023).

4     https://www.eldiario.es/madrid/somos/chamberi/opera-desde-el-balcon-para-amenizar-el-confinamiento-en-san-bernardo_1_6407560.html (accessed on 2 October 2023).

5     See note 1 above.

6     See note 3 above.

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
