# Peer review of "Sharing Art as a Daily Resistance Strategy in Madrid during the 2020 Lockdown: 50 Days of Collective Experience at the Plaza de San Bernardo"

_socsci, doi:10.3390/socsci12110608_

Round 1
Reviewer 1 Report
Overall, this research explores how individuals and communities used art as a means of resistance and well-being during times of crisis, particularly during the early months of the COVID-19 pandemic. The study focuses on a specific case in Spain, where the Plaza de San Bernardo community engaged in live artistic interactions from balconies.
Key Points and Main Arguments:
-
The study confirms the findings of previous research on the significant role played by live interactions from balconies in Spain during the lockdown. This includes activities such as the daily applause for healthcare workers, bingo games, and musical events, which fostered neighborly interactions and a sense of community.
-
The Plaza de San Bernardo community, while not a representative sample of the Spanish population, experienced a profoundly positive impact on communal engagements during the crisis. This exemplified how crises can bring communities closer together.
-
The study highlights the transformation of neighborly interactions from limited pre-crisis engagement to intense and supportive community involvement during the pandemic.
-
The research underscores the direct association between the effects of communal applause and artistic balcony activities, emphasizing the positive impact of these activities on social cohesion and emotional connection among participants.
-
There is a call for further research to comprehensively assess the evolution of communal applause traditions in Spain, especially with regard to changes in perception due to the introduction of political dimensions.
-
The study aligns with previous research indicating that musical activities were highly beneficial for enhancing well-being and emotional resilience during the lockdown. It emphasizes the significance of music and live artistic activities in fostering emotional resistance and well-being.
-
Future research is encouraged to explore how communities collectively seek and exchange effective methods and content for navigating crises through live communal experiences.
-
Longitudinal research on the ongoing development of these practices during social crises is suggested to provide a deeper understanding of their impact and evolution.
I found that both the literature review and the introductions were missing the linkage to art therapy literature that deals specifically with resilience and creativity and how art-making and art engagements support adaptation to socio-political strife (including COVID-19). Here are some suggested papers to consider exploring and possibly integrating in order to strengthen the background as well as the applicable outcomes of the paper:
Miller, G., & McDonald, A. (2020). Online art therapy during the COVID-19 pandemic. International journal of art therapy, 25(4), 159-160.
Feen-Calligan, H., Grasser, L. R., Smigels, J., McCabe, N., Kremer, B., Al-Zuwayyin, A., ... & Javanbakht, A. (2023). Creating through COVID: Virtual art therapy for youth resettled as refugees. Art Therapy, 40(1), 22-30.
Usiskin, M., & Lloyd, B. (2020). Lifeline, frontline, online: adapting art therapy for social engagement across borders. International Journal of Art Therapy, 25(4), 183-191.
Kinnear, S. (2023). ‘There is a sense of bravery in having to make a mark’: Resilience and art therapy in South Africa. South African Journal of Arts Therapies, 1(1), 50-71.
Metzl, E. S. (2009). The role of creative thinking in resilience after hurricane Katrina. Psychology of Aesthetics, Creativity, and the Arts, 3(2), 112.
Bender, B., Metzl, E. S., Selman, T., Gloger, D., & Moreno, N. (2015). Creative soups for the soul: Stories of community recovery in Talca, Chile, after the 2010 earthquake.
Keisari, S., Feniger-Schaal, R., Butler, J. D., Sajnani, N., Golan, N., & Orkibi, H. (2023). Loss, adaptation and growth: The experiences of creative arts therapists during the Covid-19 pandemic. The Arts in Psychotherapy, 82, 101983.
All in all, the study adds to the contributions linking creativity and resilience and could be more directly linked to the growing field of art therapy as it directly applies to the importance of art and communal engagement to enhance resilience and well-being during times of crisis.
Author Response
Dear Reviewer no. 1:
Thank you very much for your revision of the manuscript and for your helpful suggestion. I have added a new explicit linkage to art therapy during pandemic both in the introduction and the theoretical review. Among your kind suggestion of very interesting papers, Miller and McDonald and Feen-Calligan et al. are particularly meaningful for this research.
Thank you very much once more.

Reviewer 2 Report
The article focuses on assessing the social consequences of the COVID-19 pandemic and on ways of articulating groups and communities to carry out actions to promote common well-being.
According to the author, the emotional weight of COVID-19 gave way to the emergence of spontaneous artistic expressions that supported resilience and emotional self-regulation. The most common ones were articulated through online connections. But there were also "live" art-type activities that emerged unexpectedly. According to the author:
These alternative activities, with musicians often at the forefront, were not only driven by the desire for collective entertainment, encouragement, and connection but also evolved into spontaneous conduits for fostering a deeper sense of collective interaction and well-being (…) [T]hese periodic activities organized by neighbors – including small concerts, recitals for children, and open-air film projections on walls – assumed a key role in helping many citizens cope with grief and isolation (2)
The article focuses on the detailed analysis of one of these artistic activities in a neighborhood of Madrid known as La Plaza de San Bernardo. The author describes the geography of the neighborhood, its social characteristics (including the typology of its population), its centers of sociability, the emblematic spaces of daily interaction, etc. With the confinement produced by the pandemic, s/he describes the different artistic activities that began to articulate a resilient response to the isolation.
Having described the context of the research and the case study, the author clearly establishes two objectives of his/her article: to test whether these initiatives and interactions really helped neighbors (compared to neighborhoods where they did not occur) and whether they functioned in a more positive way than the internet- online communications. Based on these objectives, s/he establishes two clearly defined hypotheses:
Hypothesis 1: that live artistic activities shared in neighborhoods served as potent catalysts for cohesion, emotional resilience, and resistance during the crisis. Hypothesis 2: live interactions, even when initiated by citizens with whom there had been no prior contact until the crisis, were favored and more effective in terms of emotional closeness within the local community. (4)
The methodology used was based on the analysis of 60 in-depth personal interviews. The results have allowed the full confirmation of hypothesis 1. As for hypothesis 2, they have shown that the preference for live-interactions versus the effectiveness of interaction through media and online connections is not clear.
The article describes with extreme precision the deployment of the methodology, the analysis of the data and the reasoning of the research conclusions. This whole process reveals an interest in the rigor of the analysis of the interviews that allow to fix the results of the research.
However, I believe that the strongest part of the article lies in the meticulous interpretation of the results. The methodology is related to specific contexts that determine its scope and clearly defines the concrete role of these initiatives in various aspects, such as the organization of daily time, the spread of well-being and, above all, the role of the connection of the participants in the creation of a sense of community.
In my opinion, the article is a piece of exemplary research. It deals with a very specific topic and from it tries to draw general consequences. The concrete approach allows the application of a well-thought-out methodology based on in-depth interviews. The conclusions of these interviews are cross-checked with contextual aspects and with other alternative practices of interaction during confinement. I believe that no major issues are left unaddressed. Although it is not part of the objectives of the article, perhaps it would not be bad if the author would support with some bibliographical references some of the psychological terms used, such as "trauma" or "resilience". But this is a secondary aspect, simply useful for the reader to know his sources in these matters.
I should add that it is a pleasure to read texts that present clear objectives, adjusted to a concrete corpus and with a rigorous discussion of the results. It is, moreover, elegantly written. It moves away from the obscure jargon that dominates most academic publications and from the theoretical corsets that, in general, serve to ensure that the objects of study simply confirm the initial preconceptions. In a rare gesture of intellectual honesty, the text allows the reader to confront the data and understand why some research hypotheses are confirmed and others are not.
Author Response
Dear Reviewer no. 2:
Thank you very much for your revision of the manuscript, your attention and words are deeply encouraging. Following your helpful suggestion, I have added new references on the concepts of “trauma” and “resilience” regarding the specific case of the pandemic, both in the introduction and the theoretical review.
Thank you once more.

Author Response
Dear Reviewer no.3:
Thank you very much for your attention and remarks.
